# Tumor Mutation Burden, Expressed Neoantigens and the Immune Microenvironment in Diffuse Gliomas

**DOI:** 10.3390/cancers13236092

**Published:** 2021-12-03

**Authors:** Guangyang Yu, Ying Pang, Mythili Merchant, Chimene Kesserwan, Vineela Gangalapudi, Abdalla Abdelmaksoud, Alice Ranjan, Olga Kim, Jun S. Wei, Hsien-Chao Chou, Xinyu Wen, Sivasish Sindiri, Young K. Song, Liqiang Xi, Rosandra N. Kaplan, Terri S. Armstrong, Mark R. Gilbert, Kenneth Aldape, Javed Khan, Jing Wu

**Affiliations:** 1Neuro-Oncology Branch, Center for Cancer Research, National Cancer Institute, National Institutes of Health, Bethesda, MD 20892, USA; guangyangyu11@fudan.edu.cn (G.Y.); ying.pang@nih.gov (Y.P.); mythili.merchant@nih.gov (M.M.); alice.ranjan@nih.gov (A.R.); olga.kim@nih.gov (O.K.); terri.armstrong@nih.gov (T.S.A.); mark.gilbert@nih.gov (M.R.G.); 2Genetics Branch, Center for Cancer Research, National Cancer Institute, National Institutes of Health, Bethesda, MD 20892, USA; chimene.kesserwan@nih.gov (C.K.); vineela.gangalapudi@nih.gov (V.G.); abdalla.abdelmaksoud@nih.gov (A.A.); weij@mail.nih.gov (J.S.W.); hsien-chao.chou@nih.gov (H.-C.C.); wenxi@mail.nih.gov (X.W.); sivasish.sindiri@nih.gov (S.S.); songyo@mail.nih.gov (Y.K.S.); 3Laboratory of Pathology, Center for Cancer Research, National Cancer Institute, National Institutes of Health, Bethesda, MD 20892, USA; xil2@mail.nih.gov (L.X.); kenneth.aldape@nih.gov (K.A.); 4Pediatric Oncology Branch, Center for Cancer Research, National Cancer Institute, National Institutes of Health, Bethesda, MD 20892, USA; rosie.kaplan@nih.gov

**Keywords:** glioma, tumor mutation burden, neoantigen, immune score, germline mutation, antigen processing and presentation, immunotherapy

## Abstract

**Simple Summary:**

Tumor mutation burden (TMB) has shown promise as a biomarker for immune checkpoint blockade therapy in some cancers, but not consistently in gliomas. The goal of our study was to systematically investigate the association between TMB, expressed neoantigens, and the tumor immune microenvironment in IDH-mutant and IDH-wildtype gliomas, which are two types of biologically distinct gliomas. We demonstrated that TMB positively correlated with expressed neoantigens, but inversely correlated with immune score in IDH-wildtype tumors but showed no correlation in IDH-mutant tumors. The antigen processing and presenting (APP) score may have potential as a clinical biomarker to predict immune therapy response in gliomas. Lastly, 19% of patients had pathogenic or likely pathogenic germline mutations, primarily in DNA damage repair genes.

**Abstract:**

Background: A consistent correlation between tumor mutation burden (TMB) and tumor immune microenvironment has not been observed in gliomas as in other cancers. Methods: Driver germline and somatic mutations, TMB, neoantigen, and immune cell signatures were analyzed using whole exome sequencing (WES) and transcriptome sequencing of tumor and WES of matched germline DNA in a cohort of 66 glioma samples (44 IDH-mutant and 22 IDH-wildtype). Results: Fourteen samples revealed a hypermutator phenotype (HMP). Eight pathogenic (P) or likely pathogenic (LP) germline variants were detected in 9 (19%) patients. Six of these 8 genes were DNA damage repair genes. P/LP germline variants were found in 22% of IDH-mutant gliomas and 12.5% of IDH-wildtype gliomas (*p* = 0.7). TMB was correlated with expressed neoantigen but showed an inverse correlation with immune score (R = −0.46, *p* = 0.03) in IDH-wildtype tumors and no correlation in IDH-mutant tumors. The Antigen Processing and Presentation (APP) score correlated with immune score and was surprisingly higher in NHMP versus HMP samples in IDH-wildtype gliomas, but higher in HMP versus NHMP in IDH-mutant gliomas. Conclusion: TMB was inversely correlated with immune score in IDH-wildtype gliomas and showed no correlation in IDH-mutant tumors. APP was correlated with immune score and may be further investigated as a biomarker for response to immunotherapy in gliomas. Studies of germline variants in a larger glioma cohort are warranted.

## 1. Introduction

Gliomas are the most common primary malignant brain tumor and remain a fatal disease [1]. They are challenging to treat, largely due to the high level of intra- and inter-tumoral heterogeneity and a genomic landscape that constantly evolves due to selective pressure in response to therapies [2]. In addition, the immunosuppressive tumor microenvironment (TME) counteracts the efficacy of therapies, particularly immunotherapies [2]. 

In the past decade, immunotherapy such as immune checkpoint blockade has emerged as an effective therapeutic approach for several types of cancers, such as melanoma and lung cancer [3]. However, response to immunotherapy varies in patients with the same type of cancer, demonstrating the importance of identifying predictive biomarkers [3]. Tumor mutation burden (TMB), which is often proportional to the neoantigen burden, has emerged as a promising predictive biomarker of immune response in melanoma and lung cancer [4]. These efforts are highlighted in the KEYNOTE-158 study, which led to the recent US Food and Drug Administration (FDA) approval of using pembrolizumab, an anti-PD1 immune checkpoint inhibitor, in solid tumors with a TMB above 10 mutations per mega base (Mb) (defined as having a hypermutator phenotype (HMP) [5]. However, this correlation between TMB and response to immunotherapy has not been consistently observed in gliomas [6,7]. 

A recently published seminal study by Touat et al. comprehensively analyzed the molecular determinants of TMB in over 10,000 glioma samples [6]. Two major pathways to hypermutation were elucidated: a de novo pathway associated with constitutional defects in mismatch repair (MMR) genes, and an acquired resistance driven by MMR deficiency following temozolomide (TMZ) treatment. While MMR deficient tumors are more likely to accumulate TMB, they were found to have a lack of T cell infiltrates and a low rate of response to anti-PD1 therapy. This study provided evidence that TMZ can drive the accumulation of mutations without promoting a response to immunotherapy. While detailed characterization of the phenotypic and molecular features of hypermutated gliomas has been performed, a systematic analysis of the associations between TMB, expressed neoantigens, and tumor microenvironment has not been previously performed and may provide a better understanding of the discordance between a high TMB and poor response to immunotherapy in gliomas. 

In addition, the mechanisms underlying this discordance may not be the same in biologically distinct subsets of gliomas. Isocitrate dehydrogenase (IDH)-mutant gliomas have a distinct tumor biology compared to IDH-wildtype gliomas at genetic and epigenetic levels [8]. Moreover, *IDH* mutation status has been considered a favorable predictive biomarker for clinical outcomes [9]. The discovery of mutations in *IDH* genes has led to a better understanding of glioma biology as well as a major change in diagnostic criteria and standards of care.

In this study, we performed a comprehensive genomic analysis including whole exome sequencing (WES) and transcriptomic analysis of primary and recurrent tumor samples in both IDH-mutant and IDH-wildtype gliomas. Furthermore, we examined germline cancer predisposition genes (CPGs) by conducting WES of matched blood samples. The focus of our study was to analyze the correlations between TMB, expressed neoantigens, immune score of the tumor microenvironment, and antigen processing and presentation (APP) function in IDH-wildtype and -mutant gliomas separately. Our data shows promise for further investigating APP score as a clinical biomarker for determining immune response in glioma patients. 

## 2. Results

### 2.1. Sample Characteristics

A total of 66 tumor samples and matched blood samples collected from 48 glioma patients from January 2016 to March 2020 were analyzed. As summarized in Table 1 and further expanded on in Appendix A, the sample cohort included both IDH-mutant (n = 44) and IDH-wildtype (n = 22) tumors, as well as samples collected from primary (n = 13) and recurrent disease stages (n = 53), ranging from the 1st to more than 5th recurrence, which represent different stages of the disease (Table 1, Appendix A). The samples used in this study also exhibited different histology and tumor World Health Organization (WHO) grades. 

### 2.2. Pathogenic Germline Mutations 

Among 48 patients, nine (19%) were found to carry heterozygous pathogenic (P) or likely pathogenic (LP) germline alterations in eight cancer predisposition genes (CPGs): *TP53*, *MUTYH*, *BLM*, *RET*, *ERCC6*, *MITF*, *BRIP1*, and *MSH2* (Table 2). Importantly, six of them, except for *RET* and *MITF*, are involved in the DNA damage repair (DDR) pathway, indicating the importance of genomic instability in glioma genesis. Among these nine patients, seven had IDH-mutant gliomas and two had IDH-wildtype gliomas. No correlation was found between P/LP variants and the *IDH* somatic mutation status (P/LP germline variants in 21.9% of patients with IDH-mutant gliomas versus 12.5% of patients with IDH-wildtype gliomas, two-tailed Fisher’s exact test, *p* = 0.7). Analysis of the TMB revealed HMP in two patients with IDH-mutant gliomas at the time of disease recurrence, each carrying P/LP germline variants in *MUTYH* and *ERCC6,* respectively and in one patient with de novo IDH-wildtype tumor (NCI0392) carrying a pathogenic variant in *MSH2*. Therefore, three of 11 (27.3%) patients with HMP tumors and three of 37 (7.9%) patients with NHMP tumors had P/LP germline mutations in DDR genes. However, we found no association between the hypermutation phenotype and the presence of P/LP germline variants in the DDR pathway (two-tailed Fisher’s exact, *p* = 0.12). Taken together, mutations in DDR-related genes are common among P/LP germline variants. However, the association of DDR germline variants with HMP development needs to be further studied in a larger cohort. 

### 2.3. Mutational Landscape 

Among the 66 samples, tumor DNA was available for 65 samples. A total of 28,630 high confidence somatic mutations were detected by WES analysis. Using our in-house tiering system, 353 pathogenic or hotspot mutations (tier 1) were detected [11]. The top somatic mutations in the tier 1 list are summarized in Figure 1A. The common genetic alterations include *IDH1* (58%), *TP53* (58%), *ATRX* (47%), *IDH2* (11%), CIC (13%), *SETD2* (13%), *PIK3CA* (11%), *PIK3R1*(11%), *PTEN* (8%), and *RB1*(8%), which are consistent with previously reported genomic alterations in gliomas [12]. To further understand the potential of these high confidence somatic mutations to generate tumor antigens, we examined the percentage of mutant genes that are expressed. We looked for the exact variant reads from RNAseq of the corresponding tumor using a set of filters (VAF ≥ 0.1, total RNA coverage ≥ 10, variant coverage ≥ 2) to identify the expressed somatic mutations from all high confidence somatic mutations. Tier 1 mutations were more likely to be expressed compared to all high confidence somatic mutations (52.4% vs. 30.1%, *p* < 0.0001; Fisher’s exact test, two-tailed) (Figure 1B). 

A comparison of the genetic alterations in recurrent IDH-mutant gliomas with those in the matched newly diagnosed tumors demonstrated a significant number of acquired mutations that are specific to the recurrent tumors [13]. To examine the genetic alterations that evolve through disease progression, we analyzed the high confidence somatic mutations in the samples collected at early recurrences to their matched samples collected at later recurrences. Patients CL0046 and CL0301, who developed HMP, had the highest number of shared mutations in the matched samples (Figure 1C). This suggests that recurrent HMP gliomas harbor mutations that persist, indicating the existence of a resistant clone. Patient CL0238, previously reported by our group to harbor a pathogenic fusion gene *BCR-ABL* [14], was diagnosed with a NHMP glioma that also had a high percentage of shared mutations, indicating that the fusion event of *BCR-ABL* occurred early and that it is an oncogenic driver leading to rapid progression of disease without significant clonal divergence. 

To better understand the mutational profiles of gliomas, we calculated the TMB in our sample cohort using WES data of tumor samples and their matched blood samples. The TMB in all 65 samples ranged from 0.6 to 254 mutations per Mb. We then compared TMB in newly diagnosed (ND) and recurrent tumors in both IDH-wildtype and IDH-mutant gliomas. TMB values of recurrent samples were significantly higher than that of ND samples for IDH-mutant tumors (median: 1.17 versus 2.63, *p* = 0.007). However, no statistically significant difference in TMB values was found between recurrent and ND samples for IDH-wildtype tumors (Figure 2A). Using 10 mutations per Mb as the cutoff, 14 tumor samples were defined as HMP and 51 samples were NHMP. Among the 14 HMP samples, 11 samples were IDH-mutant and three were IDH-wildtype, and among the 51 NHMP samples, 32 were IDH-mutant and 19 were IDH-wildtype (Appendix A). There was no difference in hypermutation phenotype incidence between IDH-mutant and IDH-wildtype gliomas (26% and 13.6%, respectively, two-tailed Fisher’s exact test, *p* = 0.35).

The TMZ-induced mutational signature (G:C > A:T), defined as signature 11, is often observed in post-TMZ recurrent gliomas and relevant to clinical management of glioma patient [6]. Touat et al. demonstrated that over 98% of post-treatment HMP gliomas showed signature 11 and that exposing MMR-deficient cells to TMZ induces HMP with signature 11, suggesting that HMP and signature 11 represent MMR deficiency and TMZ resistance [6]. In order to examine the prevalence of signature 11 in all tumors exposed to TMZ, we analyzed the 52 samples that were collected at disease recurrence in our cohort. Among all recurrent samples, 43 were from tumors that had prior exposure to TMZ or TMZ + radiotherapy (Table 1), and 42 of them had DNA samples. Among all samples, 15 of them demonstrated signature 11. Interestingly, 35.7% (15 of 42) samples exposed to TMZ developed signature 11, and 93.3% (14 of 15) were IDH-mutant gliomas. Of the other 27 samples without signature 11, 16 were IDH-mutant and 11 were IDH-wildtype. With the exposure to TMZ, 45.2% (14 out of 30) IDH-mutant tumors and 8.3% (1 out of 12) IDH-wildtype tumors developed signature 11, suggesting that the IDH-mutant tumors were more likely to harbor signature 11 following TMZ exposure (two-tailed Fisher’s exact test, *p* = 0.02).

### 2.4. TMB, Neoantigens, and Immune Signatures

Tumor neoantigens play a vital role in anti-tumor immunity. To better understand the immune landscape of gliomas, neoantigens from tumor samples in our cohort were predicted from mutations detected by WES of tumor DNA. In total, we found 1963 neoantigens (derived from 1325, 4.6% of all high confidence somatic variants) predicted to have a high binding affinity to human leukocyte antigen I (HLA-I) (IC_50_ < 500 nanomolar (nM)) and a lower HLA-I binding affinity (IC_50_ > 500 nM) to the corresponding wildtype peptides. Since immune cells must recognize neoantigens that are expressed and presented by HLA molecules on the tumor cell surface, we filtered out 619 expressed neoantigens from the predicted neoantigens by using a cut off total RNA read coverage ≥ 10, matched variant RNA read coverage ≥ 2 and VAF ≥ 0.1 (Appendix A) (31.5%, 619/1963). As fusion genes are also a source of neoantigens, fusion gene-derived neoantigens were also included in the neoantigen calculation. In our samples, 20 high-confidence fusion gene-derived neoantigens were detected (IC_50_ < 500 nM). While the predicted neoantigens are directly derived from somatic mutations and are expected to correlate with TMB, we confirmed that the expressed neoantigens also have a statistically significant correlation with TMB in all samples (Pearson R = 0.52, *p* < 0.0001) (Figure 2B). 

Although a correlation between TMB and the tumor immune response has been reported in other cancers, there is a discordance in gliomas [15]. Given the overall strong correlation between TMB and the expressed neoantigens, we next examined the correlation between TMB and tumor immune scores. In our cohort, TMB showed an inverse correlation with immune score in IDH-wildtype samples (R = −0.46, *p* = 0.03) (Figure 2C), and no correlation in IDH-mutant gliomas (Figure 2D), suggesting that the *IDH* mutation has an impact on the correlation of TMB and immune score. 

To characterize the tumor immune microenvironment of HMP and NHMP in IDH-mutant and IDH-wildtype tumors, we performed ssGSEA using the transcriptomic data that was available for 60 samples in our sample cohort. Immune cell specific gene sets were used to calculate enrichment scores for infiltrating immune cell types and describe overall “immune signature score” in each sample [16]. Most of the immune cell infiltration scores for CD8 T cells, CD4 T cells, subtypes of dendritic cells, and macrophages were higher in IDH-wildtype samples compared to IDH-mutant samples (Figure 2E and Appendix A), which is similar to previous findings in primary gliomas from The Cancer Genome Atlas (TCGA) dataset [17]. It was also notable that several subsets of T cells and NK cells had a higher score in NHMP compared to HMP in IDH-wildtype tumors. However, no significant difference was observed between HMP and NHMP in IDH-mutant tumors (Appendix A). Overall, the immune signature clustered better by *IDH* mutation status, IDH-wildtype versus IDH-mutant, than by TMB, HMP versus NHMP (Figure 2E). 

In order to better understand the immune signatures of the tumor microenvironment, we examined the infiltrating immune cell subtypes inferred by CIBERSORT scores [16]. As shown in Appendix A, regardless of *IDH* status or TMB, all glioma groups showed similarly high percentage of immune cells classified as M2 macrophages, but no significant difference between groups (one-way ANOVA test, *p* = 0.78) (Appendix A). Monocytes and activated mast cells also had relatively high percentages (total average 13.9% and 12.7%, respectively) of infiltration compared to other immune cells such as CD8 T cells (total average 3.5%). Furthermore, there was no significant difference in CD8 T cell infiltration between HMP and NHMP samples, irrespective of *IDH* mutation status (One-way ANOVA test, *p* = 0.28) (Appendix A). These data are consistent with previous findings that M2 macrophages are the dominant immune cell in the glioma microenvironment, whereas CD8 T cells are a minority [15]. In addition, the similar proportions of these immune cells across all groups are unlikely to explain the different correlations of TMB and immune scores in IDH-mutant and IDH-wildtype gliomas. 

### 2.5. Antigen Processing and Presentation

Effective immune responses against tumors largely depend on immune cells recognizing antigens presented on the tumor surface. HLA-I loss and defects in the antigen processing machinery were reported to be common in various cancers, including gliomas [18,19,20,21]. To assess the ability of antigen presentation in gliomas, we first explored the expression of the major histocompatibility complex class I. As shown in Figure 3A, no significant difference in the expression levels of HLA-A, B, or C was found between HMP and NHMP samples in either IDH-mutant or IDH-wildtype tumors. These results suggest that HLA expression is unlikely to be the cause of the different correlation of TMB and immune scores in IDH-mutant and IDH-wildtype gliomas. 

To further understand the discordance between neoantigen burden and immune infiltrate function in the tumor microenvironment, the KEGG Antigen Processing and Presentation (APP) score between HMP and NHMP samples was compared in both IDH-mutant and -wildtype gliomas. As shown in Figure 3B, the KEGG APP score was significantly higher in NHMP samples compared to HMP samples in IDH-wildtype tumor (median 0.2385 versus −1.518, *p* = 0.014). In contrast, a significantly higher KEGG APP score was found in HMP samples versus NHMP samples in IDH-mutant gliomas (median 0.35 versus −0.39, *p* = 0.03). To better understand the effect of APP score on the tumor microenvironment, the correlation between APP score and immune score was analyzed. As shown in Figure 3C, the APP score had a statistically significant correlation with immune score in gliomas (R = 0.45, *p* = 0.0003). These data indicate that APP function is different between HMP and NHMP samples with different *IDH* mutation status but correlates with immune score in our sample cohort.

### 2.6. Immunosuppressive Gene Expression in Gliomas

To understand the role of immunosuppressive factors in the tumor microenvironment of HMP and NHMP gliomas, we analyzed the expression of well-known immunosuppressive genes. The expression levels of most examined immunosuppressive genes did not show significant differences between HMP and NHMP samples in IDH-mutant gliomas, except for *TGFB1*, which had a trend of higher expression in NHMP IDH-mutant samples (median 4.34 versus 2.9, *p* = 0.054) (Appendix A). In IDH-wildtype gliomas, the immunosuppressive genes that showed a statistically significant difference in expression between NHMP and HMP samples were *PD1* and *PDL1* (median *PD1*: 0.19 versus 0.08, *p* = 0.04; *PDL1*: 1.06 versus 0.27, *p* = 0.04) (Figure 3D). These data suggest a potential therapeutic role of targeting *TGFB1* and *PD1/PDL1* in IDH-mutant and IDH-wildtype gliomas, respectively. 

## 3. Discussion

TMB has been used as a predictive biomarker of response to immune checkpoint blockade therapy in several cancers, including melanoma and lung cancer [4]. However, a correlation between TMB and response to immunotherapy has not been observed in gliomas consistently [6]. In this study, we focused on a systematic assessment of the TMB, expressed neoantigens, and the tumor immune microenvironment in both IDH-wildtype and IDH-mutant gliomas, which have distinct tumor biology. Compared to IDH-wildtype glioma, IDH-mutant gliomas were more likely to accumulate mutation burden during their disease progression and more likely to harbor signature 11 following the exposure to TMZ. Most importantly, while TMB had a positive correlation with expressed neoantigens, it showed an inverse correlation with immune scores in IDH-wildtype gliomas and no correlation in IDH-mutant gliomas. In addition, we found a significantly higher APP score in NHMP compared to HMP samples in IDH-wildtype gliomas, but a higher APP score in HMP compared to NHMP in IDH-mutant gliomas. Together with the strong correlation between APP score and immune score, the data suggests that APP score could be further investigated as a biomarker for predicting response to immunotherapy, and that the impact of TMB on the immune signature depends on the *IDH* mutation status. Finally, we also analyzed germline alterations of CPGs, particularly P/LP genes, and explored the correlation with other tumor driver genes, such as *IDH* and *TP53*. Our results provide evidence for further evaluating P/LP germline variants in a larger glioma cohort and a potential value in screening patients prior to receiving treatment.

### 3.1. Germline Variants of P/LP CPGs in Gliomas

Despite the fact that we only analyzed a small cohort of glioma patients, 19% of them carried a germline monoallelic P/LP variant in CPG. The prevalence in our cohort is higher than what is reported in the literature in both pediatric and adult cancer patients [22,23]. To understand the spectrum of CPGs, particularly the P/LP mutations of CPGs in IDH-mutant and IDH-wildtype gliomas, we collected and analyzed germline genomic information in all cases. Although there was no statistically significant association between *IDH* mutation status and the occurrence of P/LP mutations, interesting observations between CPGs and somatic variants in the tumors were made. First, a germline *TP53* mutation (p.R209Q) was detected in a patient with grade 3 astrocytoma (OM161). In addition to *TP53* mutation and loss of heterozygosity (LOH), a frame-shift deletion of *ATRX* and somatic *IDH1* mutation, which is considered a tumor driver gene in gliomas, were also detected in the tumor. *IDH* mutation was also detected in two patients with grade 4 astrocytoma (CL0095 and CL0332) who carried a monoallelic germline mutation of *MUTYH* (p.G396D), which is a common mutation in MUTYH-associated polyposis (MAP) with an autosomal recessive inheritance [24]. Another IDH-mutant grade 4 astrocytoma patient (CL0101) was found to have a monoallelic pathogenic nonsense *BLM* mutation (p.Q548X). Biallelic *BLM* mutation usually occurs in Bloom syndrome, which features abnormal DNA repair and high levels of chromosome breaks and rearrangements [25]. Evidently, *IDH* mutations frequently occurred in patients carrying P/LP germline mutations in our patient cohort. These observations raise a question about the role of another cancer driver gene such as *IDH* mutation in the presence of germline drivers such as *TP53* mutation. Thus, it would be interesting to review the P/LP germline mutations of CPGs in a large cohort of IDH-mutant tumors. However, based on our available data, it may not be possible to determine with certainty which P/LP variants are incidental and therefore, less likely to contribute to the primary tumor diagnosis. For instance, while the *TP53* variant reported in patient OM161 is likely causal of the patient’s astrocytoma, it would be less likely for a monoallelic *MUTYH* in patients CL0095 or CL00332 to contribute to their respective tumor diagnoses.

### 3.2. TMB, Immune Signatures, and IDH Mutation Status 

There has been increasing evidence that TMB does not consistently correlate with immune response in gliomas [6,15] Our analysis revealed that TMB and immune scores are correlated differently in IDH-mutant and -wildtype gliomas. As summarized in Figure 4, in IDH-wildtype tumors, NHMP tumors have better APP function and immune scores than HMP tumors. In contrast, HMP tumors have higher APP function than NHMP, but not a better immune score, in IDH-mutant gliomas. Furthermore, APP score strongly correlates with immune score in all gliomas. While one would expect a higher TMB to result in a higher number of expressed neoantigens, in turn increasing the immune response, our findings revealed the opposite in the case of IDH-wildtype gliomas. A similar finding was described by Gromeier et al., who reported that IDH-wildtype samples with a lower TMB had higher immune inflammation, which was explained by the mechanism of neoantigen depletion/immunoediting [26]. Of note, we also found that the CD8 T cells showed a non-significant trend of being suppressed alongside APP suppression in the IDH-wildtype HMP gliomas (Appendix A), potentially dampening the immune response. Interestingly, while APP scores were elevated in the HMP subgroup in IDH-mutant gliomas, no significant correlation between the immune signature and TMB in this subset of patients was revealed. Therefore, it is possible that despite the high APP score in IDH-mutant HMP gliomas, 2-hydroxygluarate (2-HG) induces T cell suppression in some capacity. The production of this oncometabolite is a unique feature of IDH-mutant gliomas and has been previously shown to impair T cell activation and reduce T cell migration to the tumor site. Importantly, our ssGSEA data supported this because comparison of the CD8 T cell score in IDH-wildtype and -mutant gliomas revealed a significant suppression of these immune cells in the latter (Appendix A), consistent with findings from other studies [27].

### 3.3. Clinical Implications, Prospectives and Limitations

While conferring tumorigenesis, P/LP germline mutations may also provide important applications to aid patient management. For example, previous studies have shown that patients who are *MSH6* mutation carriers should avoid treatment with alkylating agents such as TMZ [28]. In our patient cohort, we detected a *MSH2* germline mutation in an IDH-wildtype glioblastoma patient (NCI0392), who was diagnosed with Lynch syndrome and had a *de novo* HMP brain tumor. In this case, alkylating agents such as TMZ should have been avoided if more treatment options were available to the patient. A patient with an IDH-mutant grade 4 astrocytoma (CL0301) was found to have a germline mutation in *ERCC6*, an important gene in the DNA double-stranded breaks (DSBs) repair pathway. This patient received more than 24 cycles of TMZ after the initial diagnosis of a lower grade astrocytoma, and both tumors from later recurrences were found to be HMP, harboring 2200 high confidence somatic mutations, which indicated a likely pathogenic function of this mutation and a potential role in HMP development when treated with TMZ. Although the association between the hypermutation phenotype and the presence of P/LP germline variants in the DDR pathway was not found to be statistically significant, investigation in a larger patient cohort is needed. Screening of those germline mutations of CPGs may provide insights to assist the clinical management of cancer patients.

In addition, further studying potential clinical biomarkers is vital for selecting patients who will benefit from immunotherapy. For an expressed neoantigen to elicit an immune response, a high APP score and HLA expression level are necessary. Currently, we do not completely understand why IDH-wildtype HMP gliomas show a decreased APP score. It is possible that critical genes are mutated at the time of development of HMP in IDH-WT tumors that disrupts APP and thus cause resistance to immunotherapy. This potential resistance mechanism can be further explored in a longitudinal study where matched tumor samples are collected and analyzed. Our findings of the correlation between APP function and immune score support testing the use of immunotherapy at an early stage of the disease for IDH-wildtype glioma patients when the TMB is low and APP function is high in a larger cohort study. The anti-PD1/PDL1 therapies may be valuable because of the increased expression level of PD1/PDL1 when TMB is relatively lower in the IDH-wildtype tumors. Interestingly, in the case of IDH-mutant gliomas, an opposite trend is seen, wherein a high APP score is seen in HMP gliomas, suggesting a potential value in considering IDH-mutant HMP gliomas for immunotherapy rather than their NHMP counterparts (Figure 4). While the findings of our study expand the knowledge of TMB, expressed neoantigens, and the tumor immune microenvironment and provided insights for clinical investigations, certain limitations are present. The conclusions are drawn from bioinformatic analyses of a sample cohort from a single institution. Further validation using in vitro and in vivo glioma models as well as larger cohort studies are thus warranted. Due to the retrospective nature of the study, the percentage of IDH-mutant glioma may not be representative of the incidence in the entire malignant glioma population. Nevertheless, our ongoing clinical trial (NCT 03718767) will provide prospectively collected data to further elucidate the correlation between TMB, expressed neoantigens, and tumor immune signatures.

## 4. Conclusions

TMB was inversely correlated with immune score in IDH-wildtype and showed no correlation in IDH-mutant gliomas. APP was correlated with immune score and may be further investigated as a biomarker for response to immunotherapy in gliomas. Studies of germline variants in a larger glioma cohort are warranted.

## 5. Materials and Methods

### 5.1. Patients and Samples

Adult patients with primary malignant brain tumors, who were evaluated at the Neuro-Oncology Branch, Center for Cancer Research, National Cancer Institute (NCI), were enrolled in NCI 16-C-0151 (NCT02851706), NCI 19-C-0006 (NCT03718767), and NCI 10-C-0086 (NCT01109394). The protocols were approved by the Institutional Review Board of the National Institutes of Health. Written consents were obtained from all patients. Both matched whole blood and brain tumor samples were collected and analyzed using the ClinOmics platform, a clinical next-generation sequencing program at NCI [29]. Tumor samples were only collected for sequencing if sufficient tissue for clinical diagnosis was available. The schema of overall experimental approach for this study is summarized in Appendix A.

### 5.2. mRNA Sequencing (RNAseq)

Tumor RNA was extracted from Formalin Fixed Paraffin-Embedded (FFPE) tumor sections by the Rneasy FFPE kit (Qiagen, Germantown, MD, USA). RNA libraries were prepared by using Illumina TruSeq RNA Access Library Preparation Kit according to the manufacturer’s protocol (TruSeq RNA Exome kits; Illumina, San Diego, CA, USA). The sequencing was performed on Illumina NextSeq500 (Illumina) according to the manufacturer’s protocols. Samples were sequenced at a depth of 40 million reads per sample. All the RNAseq data was processed by using an RNAseq data analysis pipeline, where reads were mapped to the ENSEMBL human genome GRCh37 build 71 using STAR. Single-sample Gene Set Enrichment Analysis (ssGSEA) was used for the generation of immune cell infiltration scores, immune scores, and antigen processing and presentation scores based on the previously published gene sets [16,30]. CIBERSORT was used to analyze the proportions of immune cells [29]. 

### 5.3. Whole Exome Sequencing

Tumor DNA was extracted from FFPE samples. Genomic DNA, which was used as germline exome sequencing, was extracted from the peripheral blood cells of individual patients. The exome was enriched by using SureSelect Clinical Research Exome Kits according to the manufacturer’s instructions (Agilent, Santa Clara, CA, USA). The prepared samples were sequenced on Illumina NextSeq500 (Illumina). Reportable germline mutations, which is defined as actionable genomic alterations to be targeted by the FDA approved drugs or clinical trials, were filtered out by in-house criteria [29]. TMB was defined as the number of somatic mutations in the coding region per Mb, which contain single nucleotide variants (SNVs), small insertions and deletions (INDELs) (usually less than 20 bases). TMB was calculated as indicated in the previous report [31].

### 5.4. Identification of Somatic Mutation

The bcl files of exome sequencing were converted to FASTQ files by using the bcl2fastq tool in CASAVA (Illumina). The sequences were then mapped to the human reference genome GRCH37 by using a customized NCI ClinOmics Bioinformatic Pipeline v3.2. MuTect and Strelka were used for somatic single nucleotide variant (SNV) and small indel calling respectively. The Genome Analysis Toolkit (GATK) and HaplotypeCaller (HAPLOC) for germline SNV and indel callings as previously described. High confidence somatic mutations were called by using the cutoffs: (1) tumor total coverage ≥20×, (2) normal total coverage ≥20× and (3) variant allele frequency (VAF) ≥0.10. Using these parameters, our assay has a high sensitivity of 100% and a positive predictive value (PPV) of 90% for the exome sequencing.

### 5.5. Neoantigen Prediction from Mutations and Fusions, and Expressed Neoantigen Computation

The high confidence of somatic mutations was used for the neoantigen prediction according to the previous report [32]. The amino acid change and the transcript peptide sequence were annotated by seq2HLA v2.2, HLAminer_v1.3.1, in-house developed script consensusHLA.pl, consencusSomaticsVCF, pl, VEP v.86, pvacseqtools 1.3.5. NeoFuse v1.1.1 was used for the prediction of fusion neoantigens. NeoFuse internally runs OptiType for genotyping of class-1 HLA and Arriba for predicting of fusion peptides and MHC flurry for binding affinity prediction. The neoantigen candidates with a mutant HLA type I binding score (IC_50_) lower than 500 nM, and a corresponding wild type binding IC_50_ of greater than 500 nM were selected as predicted neoantigens. The expressed high confidence neoantigens from somatic mutations were called based on the high confidence neoantigens from somatic mutations by further using the cutoffs: (1) total RNA read coverage ≥ 10, (2) matched variant RNA read coverage ≥ 2, (3) VAF ≥ 0.1. The total expressed neoantigen load was calculated by adding the high confidence expressed neoantigen mutation and high confidence neoantigen from fusion.

### 5.6. Statistical Analysis

Wilcoxon rank sum test was used for differential analyses between two subgroups. One-way ANOVA test was used in the comparison of more than two groups. Categorical variables were compared using Fisher’s exact test. All statistical analyses were performed by using GraphPad Prism software (Version 8, GraphPad Software, Inc., San Diego, CA, USA). *p* value < 0.05 was considered significant (*, *p* < 0.05; **, *p* < 0.01; ***, *p* < 0.001). 

### 5.7. Data Availability

All Data has been deposited in dbGaP and RNAseq and Somatic Data is available on an online database (https://clinomics.ccr.cancer.gov/clinomics/public/login, accessed date: 20 November 2021).

## Figures and Tables

**Figure 1 cancers-13-06092-f001:**
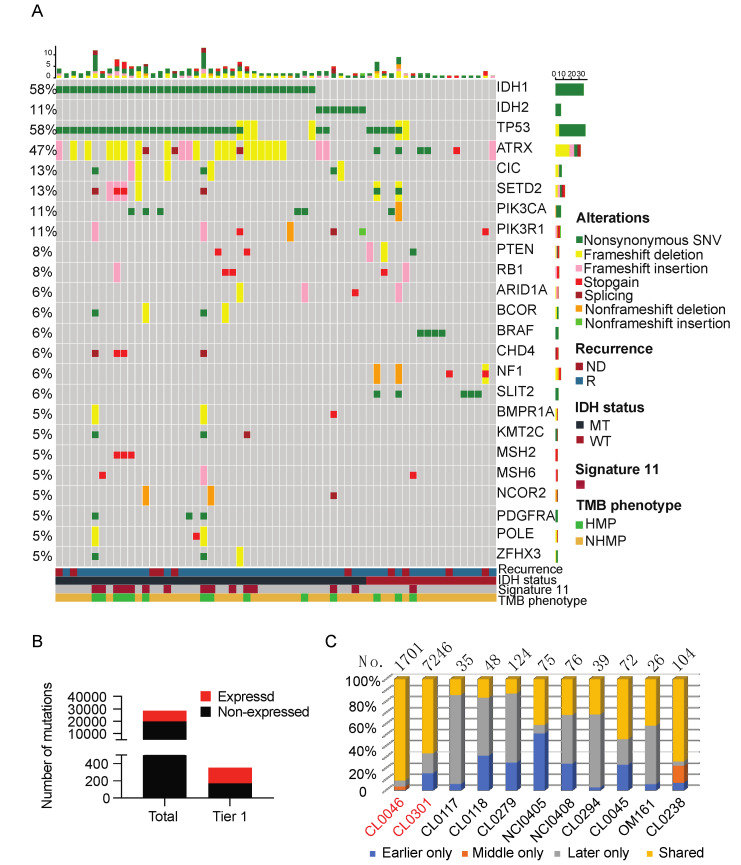
Somatic mutations detected in the sample cohort. (**A**) An integrated analysis of the sample cohort (66 samples) depicts the top tier 1 mutations. The samples are grouped by recurrence status, *IDH* mutation status, presence of mutational signature 11, and TMB phenotype. Complete information of all genetic alterations can be found in the database (https://clinomics.ccr.cancer.gov/clinomics/public/login accessed date: 20 November 2021) (**B**) High confidence somatic variants count analysis shows that tier 1 high confident somatic mutations contain a higher percentage of expressed mutations than the total high confident somatic mutations. (**C**) Matched recurrent glioma samples share expressed somatic mutations. Total number of expressed somatic mutations is labeled for each patient. Patients labeled in red carry HMP tumors, and patients labeled in black carry NHMP tumors. NHMP, TMB less than 10 mutations per Mb. HMP, TMB more than 10 mutations per Mb. ND, newly diagnosed tumor. R, recurrent tumor.

**Figure 2 cancers-13-06092-f002:**
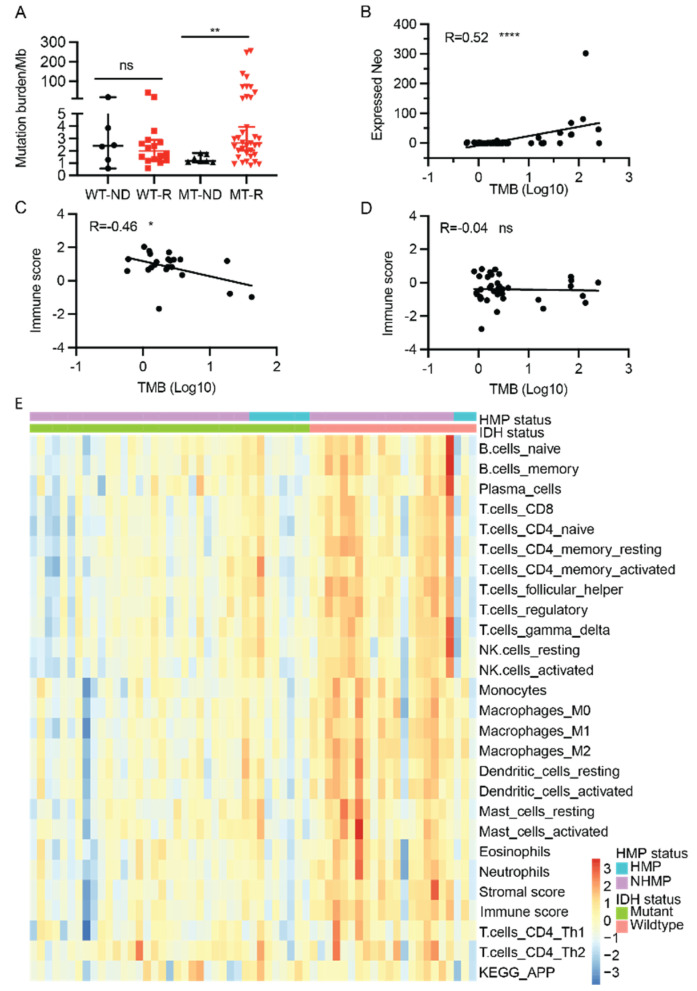
Neoantigen profile and immune signatures (n = 60). (**A**) Tumor mutation burden at initial diagnosis and recurrence in IDH-mutant (MT) and IDH-wildtype (WT) gliomas. (**B**) A significant correlation between expressed neoantigens and tumor mutation burden in all samples (R = 0.52, *p* < 0.0001. (**C**) An inverse correlation between TMB and immune score in IDH-wildtype glioma samples (R = −0.46, *p* < 0.05). (**D**) No correlation between TMB and immune score in IDH-mutant samples (R = 0.04, *p* > 0.05). (**E**) Heatmap of immune signatures in gliomas. Samples are grouped by their IDH mutation status and HMP status. Expressed neo, expressed neoantigen. ns: not statistically significant, *, *p* < 0.05; **, *p* < 0.01; ****, *p* < 0.0001. WT-ND, IDH-wildtype, newly diagnosed tumor. WT-R, IDH-wildtype, recurrent tumor. MT-ND, IDH-mutant, newly diagnosed tumor. MT-R, IDH-mutant, recurrent tumor. Wilcoxon rank sum test, ns: not statistically significant; *, *p* < 0.05.

**Figure 3 cancers-13-06092-f003:**
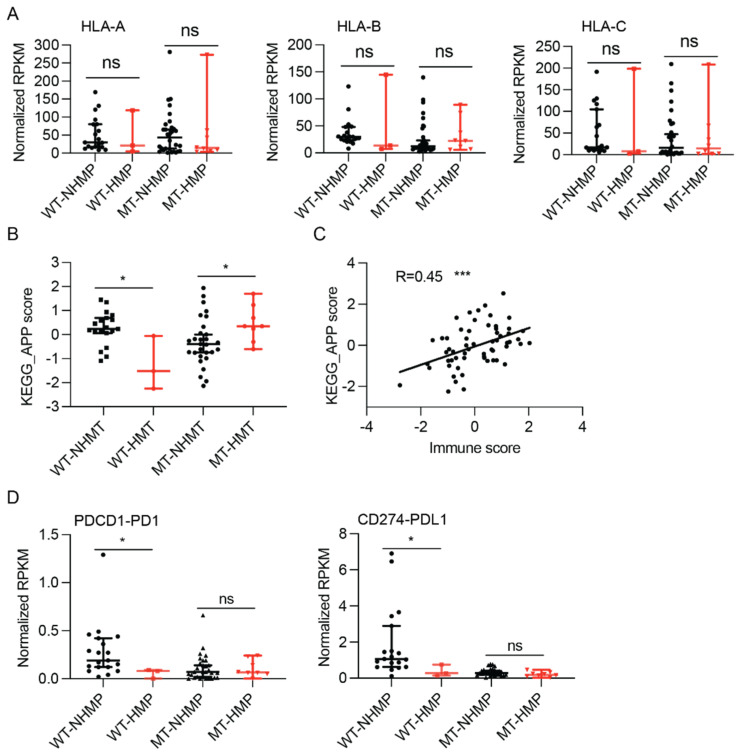
Antigen processing and presentation (APP) and immunosuppressive gene expression in HMP and NHMP glioma samples. (**A**) No difference in expression of type I HLAs between HMP and NHMP samples is detected in either IDH-wildtype or IDH-mutant tumors. (**B**) APP score is higher in NHMP than HMP for IDH-wildtype glioma samples (*p* < 0.05), but higher in HMP than NHMP for IDH-mutant glioma samples (*p* < 0.05). (**C**) KEGG_APP score correlates with immune score (R = 0.45, *p* < 0.0001). (**D**) RNA expression level of PDCD1 and CD274 in HMP and NHMP in both IDH-mutant and wildtype gliomas. APP, antigen -processing and -presentation. RPKM, reads per kilobase of transcript per million reads mapped. Wilcoxon rank sum test, ns: not statistically significant. *, *p* < 0.05. ***, *p* < 0.001.

**Figure 4 cancers-13-06092-f004:**
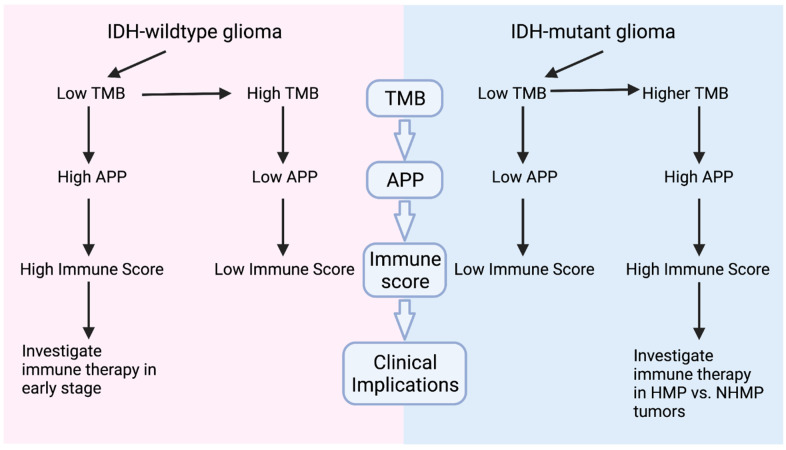
Graphic summary of study findings. In IDH-wildtype tumors, NHMP tumors have higher APP and immune scores than HMP, suggesting a need to investigate potential benefit for immunotherapy in early stage of the disease. However, in IDH-mutant gliomas, HMP tumors have higher APP scores than NHMP. Despite a lack of correlation between TMB and immune scores, investigation of immune therapy in HMP versus HNMP glioma is warranted and ongoing.

**Table 1 cancers-13-06092-t001:** Sample Characteristics by IDH mutation status.

	All Samples n = 66	IDH Mutant n = 44	IDH Wildtype n = 22
Tumor histological type			
Astrocytoma	54	32	22
Oligodendroglioma	12	12	0
WHO grade			
II	4	4	0
III	28	23	5
IV	34	17	17
Disease status			
Primary disease	13	7	6
Recurrent disease	53	37	16
No. recurrence			
0	13	7	6
1–2	33	19	14
3–5	18	16	2
>5	2	2	0
Tumor mutation burden *			
NHMP	51	32	19
HMP	14	11	3
Prior brain tumor therapies **			
TMZ/TMZ+RT	43	31	12
XRT	4	1	3
Others ***	19	12	7

Note: * Tumor mutation burden is available in 65 sample that has tumor DNA available. ** Treatments received by the patient prior to sample collection. *** Other therapies include surgical resection in newly diagnosed tumors, clinical trial therapies, and Tumor Treating Field. Abbreviations: IDH, isocitrate dehydrogenase; TMZ, temozolomide; RT, radiation therapy; HMP: hypermutator phenotype defined by more than 10 mutations per Mb.

**Table 2 cancers-13-06092-t002:** Pathogenic or likely pathogenic germline mutations detected in 9 patients.

Patient	Diagnosis *	Gene	Mutation	Associated Mendelian Disease	Mendelian Inheritance	ACMG-Based Classification [10]	HMP
OM161	Astrocytoma, IDH-mutant, WHO grade 4	*TP53*	p.R209Q	Li-Fraumeni syndrome	Autosomal Dominant	Likely pathogenic	No
CL0095	Astrocytoma, IDH-mutant, WHO grade 4	*MUTYH*	p.G396D	MUTYH associated polyposis	Autosomal Dominant	Likely pathogenic	No
CL0101	Astrocytoma, IDH-mutant, grade 4	*BLM*	p.Q548X	Bloom Syndrome	Autosomal Dominant	Pathogenic	No
CL0248	Astrocytoma, IDH-mutant, WHO grade 3	*RET*	p.K666N	Medullary thyroid carcinoma	Autosomal Dominant	Pathogenic/Likely pathogenic	No
CL0301	Astrocytoma, IDH-mutant, WHO grade 4	*ERCC6*	p.R670W	Cockayne syndrome	Autosomal Dominant	Likely pathogenic	Yes
CL0326	Astrocytoma, IDH-mutant, WHO grade 3	*MITF*	p.E419K	Susceptibility to cutaneous melanomaWaardenburg syndrome	Risk factorAutosomal Dominant	Risk factor/Likely pathogenic for cutaneous melanoma	No
CL0332	Astrocytoma, IDH-mutant, WHO grade 4	*MUTYH*	p.G396D	MUTYH associated polyposis	Autosomal Dominant	Likely pathogenic	Yes
NCI0391	Gliosarcoma, IDH-wildtype, WHO grade 4	*BRIP1*	p.T997fs	Fanconi Anemia	Autosomal Dominant	Pathogenic	No
NCI0392	Glioblastoma, IDH-wildtype, WHO grade 4	*MSH2*	c.1386+1G>A	Lynch syndrome	Autosomal Dominant	Pathogenic	Yes

Notes: * Diagnosis is based on “The Consortium to Inform Molecular and Practical Approaches to CNS Tumor Taxonomy” (cIMPCT-NOW update 6). Patients had multiple recurrence, the diagnosis in the table reflects the highest World Health Organization (WHO) grade. Abbreviations: WHO: World Health Organization; ACMG: American College of Medical Genetics; HMP: hypermutator phenotype defined by more than 10 mutations per Mb.

## Data Availability

All Data has been deposited in dbGaP and RNAseq and Somatic Data is available on an online database (https://clinomics.ccr.cancer.gov/clinomics/public/login, accessed date: 20 November 2021).

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
