# Peer review of "Tumor Mutation Burden, Expressed Neoantigens and the Immune Microenvironment in Diffuse Gliomas"

_cancers, 2021, doi:10.3390/cancers13236092_

Round 1

Reviewer 1 Report

This is an interesting manuscript showing the association between IDH status, the TMB and the immune score in glioma patients.

I have some questions that I hope that Authors could clarify.

1) While the study well clarifies differences between IDHwt and IDH mutant tumors, I could not find the differences, if any was present, among histopathological grades

2) The rate of IDH mutant high grade gliomas is very high in this study population. Since the majority were recurrent tumors, were they previous lower grade tumors with tumor progression? How did Authors select their patients? 

3) Also germline mutations are surprisingly high (and associated with IDH mutations according to authors): is there a selection bias of this samples?

4) Table 1 shows that all tumors had previous treatments, this is weird in primary cases (I would not include surgery as previous treatment as it obviously led to tumor sampling)

Author Response

This is an interesting manuscript showing the association between IDH status, the TMB and the immune score in glioma patients.

I have some questions that I hope that Authors could clarify.

1) While the study well clarifies differences between IDHwt and IDH mutant tumors, I could not find the differences, if any was present, among histopathological grades.

Response: We appreciate the reviewer’s comment. Our current study was focused on elucidating the correlation between tumor mutation burden (TMB) and tumor immune scores in these two biologically different tumor subsets. In IDH-wildtype tumor, TMB inversely correlated with immune score, while it showed no correlation in IDH-mutant gliomas. The major differences associated with the histological grade are tumor mutation burden. In IDH-mutant gliomas, the recurrent tumors, which usually are higher grade than the newly diagnosed disease harbor significantly higher mutation burden, as shown in Figure 2A. Most of the IDH-wildtype tumors are grade 4 at initial diagnosis and no significant increase in tumor mutation burden when the disease progress. This may also reflect the length of the disease process, which is usually longer in IDH-mutant gliomas than IDH-wildtype gliomas and thus more somatic mutations may be accumulated as consequences of the tumor progression and tumor treatments in the IDH-mutant gliomas.

2) The rate of IDH mutant high grade gliomas is very high in this study population. Since the majority were recurrent tumors, were they previous lower grade tumors with tumor progression? How did Authors select their patients? 

Response: We agree with reviewer’s concern about selection bias. However, this was a retrospective study based on the patient enrollment to the observational studies. As the reviewer rightly pointed out, percentage of the IDH-mutant glioma may not be representative of the incidence in the entire malignant glioma population. Among all 37 recurrent IDH-mutant gliomas, more than half of them were lower grade glioma prior to tumor progression. This limitation has been discussed and added to lines 451-452 in the Discussion section.

3) Also germline mutations are surprisingly high (and associated with IDH mutations according to authors): is there a selection bias of this samples?

Response: The reviewer is correct that the number of pathogenic/likely pathogenic germline mutations is high, and warrants publication. However, although we reported that 18.8% of patients had reportable germline variants, it is similar, albeit a little higher, to a previously reported rate of 15.7% in 1566 patients with advanced cancers (Newman S et al., 2021). Therefore, our findings should be validated in future prospective studies.

4) Table 1 shows that all tumors had previous treatments, this is weird in primary cases (I would not include surgery as previous treatment as it obviously led to tumor sampling)

Response: The reviewer brings up an excellent point. We agree that surgery has an important role in tumor sampling and in establishing pathological diagnosis. However, it also plays a role in cytoreduction as a therapeutic approach in gliomas. In order to improve clarity, we revised the annotation of Table 1 in line 116.

Reviewer 2 Report

The manuscript by described that TMB was inversely correlated with immune score in IDH-wildtype and no correlation in IDH-mutant tumors. Moreover, they described that APP was correlated with immune score and may be further investigated as a biomarker for response to immunotherapy in gliomas.

The  study is quite interestng and well-designed, however there is a first relevant issue:

What about validation of the identified somatic mutations?

Secondly, the authors must provide a schematic representation of the experimental approach used. Thirdly, in the discussion section paragaraphs have to be eliminated and the conclusions section must be placed at the end of discussion and not after materials and methods.

Collectively, the author have to better specify the focus of the study and made more clear the findings and their potential clinical impact, rewriting some parts of the introduction, discussion and conclusions.

Author Response

The manuscript by described that TMB was inversely correlated with immune score in IDH-wildtype and no correlation in IDH-mutant tumors. Moreover, they described that APP was correlated with immune score and may be further investigated as a biomarker for response to immunotherapy in gliomas.

The study is quite interesting and well-designed, however there is a first relevant issue:

What about validation of the identified somatic mutations?

Response: We appreciate the reviewer’s comment that validation of mutations is of importance. Although the sequencing was done in a research setting, the method used was the Standard Operating Procedure of a CLIA lab. For samples, we selected high confidence variants (Tumor total coverage >=20x, Normal germline total coverage >=20x, and VAF >= 0.10). Using these parameters, our assay has a high sensitivity of 100% and positive predictive value (PPV) of 90% for exome sequencing. Secondly, the mutations that are presented in Figure 1A were the top genes in tier 1, which are high-confidence mutations, based on the previously published tier system (Roper N et al., 2020). Our tiering system was built according to the reported and validated somatic variants. The top list genetic alterations are consistent with what is reported in glioma literature as well. Finally, we inspected all of the mutations reported in Figure 1A via Integrative Genomic Viewer (IGV). Therefore, we are confident that these are true positives. We have also added a comment in the methods section to emphasize our use of a CLIA pipeline in a research setting (Lines 526-527).  

Secondly, the authors must provide a schematic representation of the experimental approach used.

Response: We agree with the reviewer’s comment. The schematic illustration of the experimental approach was made and has been added as Figure S6 in the supplementary materials.

Thirdly, in the discussion section paragaraphs have to be eliminated and the conclusions section must be placed at the end of discussion and not after materials and methods.

Collectively, the author have to better specify the focus of the study and made more clear the findings and their potential clinical impact, rewriting some parts of the introduction, discussion and conclusions.

Response: We appreciate the reviewer’s careful review. We misunderstood the “Author’s instruction”, which listed a requirement of a Conclusion section after the Materials and Methods part. We have revised the Discussion section and added a Conclusion section at the end of the Discussion (lines 457-461). To clarify the focus of the study, we revised the manuscript in the Introduction (Lines 90-93) and Discussion (Lines 331-336).

Reviewer 3 Report

In this ms., Yu and Pang et. al. discovered the relation between tumor mutation burden and the immune microenvironment in diffuse gliomas. The results are interesting. I recommend it for publication in Cancers after minor revison.

  1. Is the sample size big enough to draw such a conclusion?
  2. Some prospectives are worth to be added in the conclusion.
  3. Formatting issues. IC50 to IC50; 500nM to 500 nm. Please check all.

Author Response

In this ms., Yu and Pang et. al. discovered the relation between tumor mutation burden and the immune microenvironment in diffuse gliomas. The results are interesting. I recommend it for publication in Cancers after minor revison.

1. Is the sample size big enough to draw such a conclusion?

Response: We understand the reviewer’s concern that the sample size of the study (n=66) cannot be considered as a large dataset and this limitation has been discussed in the manuscript. We believe that our study provided important findings which will need to be further validated in a larger prospective study.

2. Some prospectives are worth to be added in the conclusion.

Response: The Discussion has been revised to reflect the prospective and clinical implications.

3. Formatting issues. IC50 to IC50; 500nM to 500 nm. Please check all.

Response: We thank the reviewer for the thorough review. We have corrected the formatting issue in the manuscript. We apologize for the confusion by writing 500nM without a clear annotation. We have added “nanomolar” in the text (Line 288).

Round 2

Reviewer 2 Report

The authors address all the points. The manuscript can be accepted in its present form.